# SheetKey: Generating Touch Events by a Pattern Printed with Conductive Ink for User Authentication

Shota Yamanaka*
Yahoo Japan Corporation

Tung D. Ta†
The University of Tokyo

Kota Tsubouchi‡
Yahoo Japan Corporation

Fuminori Okuya§
The University of Tokyo

Kunihiro Kato¶
The University of Tokyo

Kenji Tsushio‖
The University of Tokyo

Yoshihiro Kawahara**
The University of Tokyo

Figure 1: Overview of SheetKey. (left) Gray discs and lines are printed with conductive ink. When a user slides a finger downwards on the user-touch part, the discs of the other side in the touch-generation part invoke touch events. (middle) A user unlocks the smartphone and (right) a secure room.

## ABSTRACT

Personal identification numbers (PINs) and grid patterns have been used for user authentication, such as for unlocking smartphones. However, they carry the risk that attackers will learn the PINs and patterns by shoulder surfing. We propose a secure authentication method called SheetKey that requires complicated and quick touch inputs that can only be accomplished with a sheet that has a pattern printed with conductive ink. Using SheetKey, users can input a complicated combination of touch events within 0.3 s by just swiping the pad of their finger on the sheet. We investigated the requirements for producing SheetKeys, *e.g.*, the optimal disc diameter for generating touch events. In a user study, 13 participants passed through authentication by using SheetKeys at success rates of 78–87%, while attackers using manual inputs had success rates of 0–27%. We also discuss the degree of complexity based on entropy and further improvements, *e.g.*, entering passwords on alphabetical keyboards.

**Index Terms:** H.5.2 [User Interfaces]: User Interfaces—Graphical user interfaces (GUI); H.5.m [Information Interfaces and Presentation]: Miscellaneous

## 1 INTRODUCTION

### 1.1 Background

Secure user authentication on mobile devices is required for many applications such as for banking and shopping, and for unlocking smart devices themselves. In the real world, entering a high-security room also requires personal authentication. Typical methods require that four- or six-digit personal identification numbers (PINs) or passwords be input or that a pattern of dots be connected on a touchscreen. However, these methods are vulnerable to shoulder surfing, *i.e.*, attackers stealthily looking at a valid user's input[1]. In addition, because fingerprints [17, 28] and finger warmth [2] remain on touchscreens, attackers can estimate the correct passwords by watching or stealing the valid user's smart device even without shoulder surfing. While various threats have been investigated recently, shoulder surfing has been recognized as a representative threat model in the secure user authentication field [9]. Therefore, in this paper, we focus mainly on shoulder surfing as a threat during PIN input.

To protect against such attacks, changing key arrangements [2,17], rotating key layouts [28], and using an additional channel such as finger force [17] have been proposed. Such methods improve security but require additional effort from users. Our goal is to provide a secure authentication method that valid users can easily use. Specifically, on the touchscreen side, the system requires that a complicated series of touches (single and multi-taps), called a *pattern*, be input in a quite short amount of time (*e.g.*, 0.3 s). Valid users can easily log in by using a sheet on which conductive ink is printed, as shown in Fig. 1, in which touch events are generated on the touchscreen in a required order. In contrast, attackers without sheets have difficulty logging in. While our proposed method, called SheetKey, requires this additional item, it has the following advantages compared with conventional widely used methods:

- SheetKey is secure against shoulder surfing, while a photo of a code displayed on a screen (*e.g.*, QR code) can be taken by an attacker.

- SheetKey provides higher entropy values than a comparable additional hardware item (here, a token generating a one-time 6-digit PIN).

- When users lose their SheetKeys or when these items are stolen by attackers, they can easily produce a SheetKey with a new

---

[1] We use *valid user* to refer to a person who logs in to a system legitimately and *attacker* to refer to a person who wants to log in illegally.

*e-mail: syamanak@yahoo-corp.jp
†e-mail: tung@akg.t.u-tokyo.ac.jp
‡e-mail: ktsubouc@yahoo-corp.jp
§e-mail: okuya23@akg.t.u-tokyo.ac.jp
¶e-mail: kkunihir@akg.t.u-tokyo.ac.jp
‖e-mail: tsushio@akg.t.u-tokyo.ac.jp
**e-mail: kawahara@akg.t.u-tokyo.ac.jp

Graphics Interface Conference 2020
28-29 May

pattern with a consumer printer and ink, while conventional hardware items for authentication (*e.g.*, IC cards and tokens) require several days to be reproduced.

Explanations of these advantages, other potential improvements to SheetKeys, and use case scenarios are given later in more detail. In this paper, we particularly discuss the technical aspects of SheetKey. Investigation into other aspects is included in our future work, including (a) measuring users' cognitive load or operational error rate in aligning a SheetKey to the correct position on a screen, (b) measuring the physical effort needed to keep a SheetKey in contact with a surface, (c) conducting usability experiments in more realistic scenarios such as measuring the time a user takes to take a SheetKey out of their pocket and attach it to a smartphone screen, and (d) interviewing users about any stresses they experienced in their everyday usage of SheetKey.

### 1.2 Contribution Statement

- We propose a secure authentication method that requires a complicated touch pattern to be input within a short period of time. With a SheetKey, valid users can easily input the correct pattern. In comparison, without a SheetKey, attackers can rarely input the correct pattern manually.

- For reproductivity of our proposed method, we measured the requirements for SheetKey, such as the optimal disc diameter for touch events (8 mm) and wire width for conveying touch (0.1 mm–0.5 mm). We found that these values are also appropriate for 13 touchscreen devices.

- The results of a user study demonstrate that SheetKey was secure against shoulder surfing. However, some participants indicated threats to SheetKey. For example, attackers can reconstruct a sheet by using photos that show a valid user using a SheetKey to input a password and reproduce the sheet with their printer using conductive ink, or attackers can steal a SheetKey and use it on their own smart devices to illegally log in. To solve these problems, we discuss several threat models other than temporal shoulder surfing and propose solutions for them.

## 2 RELATED WORK

### 2.1 Security

The main threat model for such touchscreen device usage is shoulder surfing. One simple countermeasure for preventing attackers from easily reconstructing a password is hiding input from surrounding people, *e.g.*, inputting a password behind a smartphone [8]. For smartphones equipped with a capacitive touchscreen, CapAuth [10] identifies a user when the user touches the screen with the index to pinky fingers by distinguishing, *e.g.*, finger lengths and contact sizes. This technique requires kernel modification to enable a debugging mode for accessing raw capacitive images and thus requires special literacy.

Another main threat model is the smudge attack, *i.e.*, an attacker reconstructs a password on the basis of finger oil remaining on touchscreens [4]. Smudge attacks work well for grid pattern input because the finger trajectory directly shows the correct pattern. Smudge attacks require that attackers have physical access to the touchscreens. We assume that SheetKey is effective against shoulder surfing, and at the same time, it protects from smudge attacks because valid users do not have to touch the screen with their fingers for password input.

Shoulder surfing is originally a threat model where one attacker can see a password entry once. Video recording (*i.e.*, a means by which attackers can repeatedly check passwords) [7, 32] has also been dealt with as a new threat model. In our user study, the participants tried to log in to the system as attackers while they could

see a SheetKey pattern, so this can be regarded as a situation similar to the threat of video recording.

The most relevant related method is Seyed *et al.*'s Cipher-Card [29], which shares some advantages with SheetKey. This card consists of ten electrodes on both the front and back sides, and each front electrode is wired to a single back electrode in accordance with a given rule or randomly. Therefore, when a front electrode is touched by a user, the touch event is conveyed to the back one, and thus, the touchscreen can sense the converted touch event. Because attackers do not know the rule for the electrode connections, they cannot input a correct PIN without a CipherCard. CipherCard also realizes two-factor authentication and high security. In addition, routed by hardware between wires, connections between each pair of two electrodes are reconfigurable, though a reconfigurable CipherCard is heavy.

Compared with CipherCard, a potential advantage of SheetKey would be that SheetKey can be massively produced by end users without any special knowledge; just a consumer inkjet printer and silver nano-particle ink are necessary, both of which are commercially available. For example, if an administrator permits 30 or 40 people to enter a secure room, the required number of SheetKeys can be easily produced within several minutes. Furthermore, we empirically show that SheetKeys can be used on alphabetical keyboards on smartphones, which require much denser arrangements of electrodes and wires than numeric keypads. We achieved this using an origami method for bending sheets, but it has neither been discussed or empirically shown whether CipherCard could be used for such other input modes.

### 2.2 Enhancing Interactions on Capacitive Touchscreens

Since capacitive touchscreens are low in cost to manufacture, scalable for smaller/larger sizes, and responsive to touch, recent smart devices come equipped with them as the default input device. Readers can refer to other papers (*e.g.*, [12, 31]) for the mechanism of current capacitive touchscreens. In short, when a finger touches a touchscreen, the electrical charge flowing through the embedded circuit leaks into the human body. If the amount of signal sensed by the receiver falls below a certain threshold, the touch driver judges that a touch operation has occurred.

Because the signal can leak into the body through other conductive objects (*e.g.*, metal and graphite), enhancing touch interactions by extending the touchable area to outside the touchscreen and branching a single touch into a multiples touches have been proposed [6, 11, 25]. For example, FlexTouch extends the touch sensing area of smartphones onto walls, tables, and floor mats by using copper tape and sheets printed with silver nano-particle ink [1]. This makes it possible to recognize the remaining amount of water in a cup on a table, the posture of the human body on a mat, etc. Also, increasing the size of a software keyboard for better usability has been proposed.

Inspired by the related work above, we propose SheetKey for enhancing user authentication. In particular, our main purpose of using a sheet with conductive ink is to make the manual input in the *touch-generation part* difficult, while keeping the touch operation (sliding a finger straight) easy for valid users. This allows valid users to input the correct pattern in a short amount of time, while attackers need a longer duration to do so.

## 3 SHEETKEY FOR INPUTTING COMPLICATED PATTERNS

Instead of manual touch-inputting of PINs or passwords, we propose a system that requires a complicated touch pattern to be inputted in a short amount of time, *e.g.*, 0.3 s. Such a pattern is almost impossible to manually input quickly, so this prevents attackers from illegally passing through authentication. Moreover, the system publishes

SheetKey data via a PDF or other formats to valid users, so they can print a SheetKey using conductive ink.

A user's touch is sensed by touchscreens via conductive ink as shown in Fig. 1. More specifically, when a user touches one of the discs of the *user-touch part* on a SheetKey attached on a capacitive touchscreen, the disc of the other side connected by a thin wire invokes a touch event on the touchscreen. We use this characteristic to enable a valid user to input a complicated touch pattern with a simple operation. As shown in Fig. 1, a user slides their finger downwards along the *user-touch part*, and the SheetKey converts the user's touch into the correct positions in the required order. In addition, the system requires a pattern from the first to final touch to be completed within a short time limit (*e.g.*, 0.3 s). In our implementations throughout the paper, we match the absolute positions of touch points.

We assume that attackers will rarely input the correct pattern without the sheet within the time limit. Here, we assume that attackers put their maximum potential effort into attacking; they can attack concurrently using all fingers of both hands. For example, even if the system requires a series of four taps within 0.1 s, attackers will sometimes pass through the authentication by setting four fingers above each required tap position and then touching the surface with slight time delays between successive touches. To improve security in terms of increasing the manual input difficulty for attackers, we offer the following guidelines.

1. **The time limit from the first touch to the final touch is approximately set to human's reaction time.** Control of the human hand based on visual feedback requires a certain length of time longer than 200 ms [26, 27]: approximately 260 ms [16] or 290 ms [24]. In comparison, sliding a finger on the *user-touch part* can be accomplished only with a ballistic (feedforward) finger movement. Therefore, we set the time limit to 0.3 s as a simple threshold.

2. **Changing the number of concurrent taps.** Adding multi-tapping with two or three fingers makes manual input more difficult. Changing the number of fingers and moving each finger to the correct positions takes longer than only single tapping, and thus, using multi-tapping will increase the manual input difficulty. For SheetKey, this can be achieved by branching the number of touches from one to two (or more) points.

In addition, increasing the number of touch inputs in the *touch-generation part* directly increases the entropy (discussed later). From this viewpoint, the second guideline above also improves security. Another advantage of the second guideline is that, while the number of tap points is increased by branching wires, valid users just have to pay attention to the sliding operation for the *user-touch part*.

Throughout this study, we use silver nano-particle ink [20] printed on transparent PET film [21] by a commercially available inkjet printer. A bottle of the ink costs 20,000 JPY ($\sim$ 185 USD) per 100 mL and is available online[2]. When we printed SheetKeys as shown in Fig. 3, we could produce more than 100 sheets per ink bottle; 100 JPY ($\sim$ 0.93 USD) per SheetKey. A sheet of A4-sized PET film ($297 \times 210$ mm$^2$), which is 10,000 JPY per 100 sheets and also easily available online[3], could be cut into four SheetKeys for smartphone screens, costing 25 JPY ($\sim$ 0.23 USD) per sheet. In total, a smartphone-screen-sized SheetKey costs 125 JPY, which is 1.16 USD. A SheetKey can be printed within 20 or 30 s.

---

[2]http://www.k-mpm.com/agnanoen/agnano_ink.html
[3]http://www.k-mpm.com/agnanoen/agnano_media.html

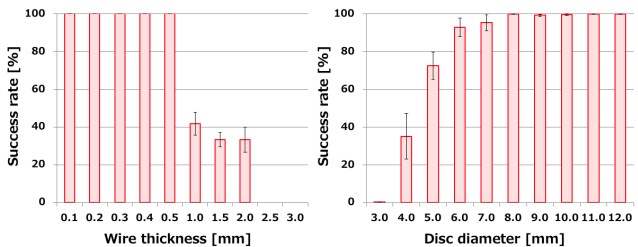

Figure 2: Recognition accuracy in pilot studies. Error bars show standard deviations across participants.

## 4 MEASUREMENT: REQUIREMENTS FOR PRINTING SHEETKEYS

SheetKeys require a conductive pattern with appropriate parameters such as an appropriate disc diameter for generating touch events and wires thin enough to not generate touch events. For example, wires that are too thick have a high density, and high-density conductive parts could generate unintended touch events. In this section, we evaluate suitable parameters for generating touch events with a high recognition accuracy.

### 4.1 Preliminary Study 1: Wire Width

#### 4.1.1 Method

First, we investigated conductive wires thin enough not to generate touch input. We used sheets that each had one conductive pattern with two discs connected with a wire of a specific width. When participants touched one disc, a touch event was generated on the other conductive disc. We used ten parameter conditions for the wire width: 0.1, 0.2, 0.3, 0.4, 0.5, 1.0, 1.5, 2.0, 2.5, and 3.0 mm. To recognize two touch input parts as different points, the distance between the centers of the two discs was set to 30.0 mm. The diameters of the two discs were fixed to 8.0 mm. We used a Surface Pro 3 ($2160 \times 1440$ pixels, $254 \times 169$ mm$^2$ of input area, 8.47 pixels/mm resolution, landscape mode). As described before, we used silver nano-particle ink [20] and a sheet of transparent PET film [21]. Three volunteers from our local university participated in this pilot study. Each participant performed tap operations 20 times for each sheet.

#### 4.1.2 Results

The results of the experiments are shown in the left graph in Fig. 2. All participants succeeded in using the sheet without generating touch input on the connection wire when the width of the connection part was set to 0.5 mm or less. When the wire width was 1.0 mm or more, unintended touch events were sometimes generated on the connection part.

### 4.2 Preliminary Study 2: Disc Diameter

#### 4.2.1 Method

We investigated the best diameter for the conductive discs to generate touch input with a high recognition accuracy. We used sheets that each had one conductive pattern with two discs of a specific diameter connected with a wire. Each disc was connected by a thin wire (0.1 mm thick). When a participant touched one conductive disc, a touch event was generated on the other disc. We used ten parameter conditions for the disc diameter: 3.0, 4.0, 5.0, 6.0, 7.0, 8.0, 9.0, 10.0, 11.0, and 12.0 mm. To recognize two touch-input points as different points, the distance between the centers of the two discs was set to 30.0 mm. Ten volunteers from our local university participated in this pilot study. Each participant performed tap operations 10 times for each sheet.

### 4.2.2 Results

We obtained the mean of the recognition accuracy for each parameter from the recorded experimental data (Fig. 2 right graph). From the results, the accuracy with diameters of 6.0 and 7.0 mm was over 90%, and for diameters of 8.0 mm or more, it was almost 100%.

We tested the sheet on 13 touchscreen devices (iPad 1, 2, and mini; iPhone 5 and 6S; Samsung Galaxy Note Edge; Galaxy Note 8; Asus Nexus 7; LG Nexus 5; Sony Vaio Pro 11 and Type T; and MS Surface Pro 1 and Pro 3) using the high recognition accuracy parameters obtained in these preliminary studies (*i.e.*, 0.5-mm-wide wire and 8-mm-diameter disc). While the touch sensors and drivers implemented in these 13 devices should be different, we assumed that the 8-mm disc diameter appropriately emulates the size of a finger touch, and thus, a high recognition accuracy was achieved.

In summary, the requirements for SheetKey are described as follows.

1. Connection wires between each conductive disc require widths of 0.1–0.5 mm to avoid touch events being generated on an unintended portion of touchscreens.

2. Conductive discs require diameters of more than 8 mm for touch events to be generated on the touch-generation part.

Therefore, for highly accurate touch-event generation, in the main study, we use a diameter of 8 mm for the discs in the *user-touch* and *touch-generation parts* and a width of 0.5 mm for the wires.

## 5 USER STUDY

The goal of this user study was to evaluate the most fundamental aspects of our proposed method. That is, whereas attackers can rarely pass through the authentication by manual touch input without using any items, valid users can more easily do so by using a SheetKey.

### 5.1 Task

This user study was divided into *valid user* and *attacker* sessions.

- **Valid user** session: Valid users used SheetKeys to pass through authentication legitimately. We measured the success rate at which valid users correctly passed through authentication by using a SheetKey. In this study, because we focused on whether users could rapidly and correctly input the pattern by using SheetKey, the experimenter pre-attached a SheetKey to the smartphone. Other factors were eliminated from this study, such as "How precisely can users attach the SheetKey onto the correct position of the touchscreen?" Otherwise, if users had attached a SheetKey at an incorrect angle on a screen, the generated touch points would not have been on the positions required by the system, and thus, all the tap trials would have failed. For this reason, we pre-attached the SheetKey to the correct position on the screen.

- **Attacker** session: Attackers wanted to illegally pass through authentication by manually inputting the pattern. We measured how often attackers correctly passed through authentication without using a SheetKey under the condition that they could always look at the SheetKey used in the **valid user** session. This simulates the situation in which attackers shoulder-surf valid users' input with SheetKeys. Similar to the condition for *valid users*, other factors in realistic shoulder surfing were eliminated such as "Can attackers see the correct input pattern even when valid users try to hide the screen?" Therefore, SheetKeys were put on a table, so attackers could always see them.

Table 1: Experimental apparatus

| Smartphone | LG Nexus 5 [19] |
|---|---|
| Android version | Android 6.0 Marshmallow [13] |
| Conductive ink | NBSIJ-MU01 [20] |
| Substrate | NB-TP-3GU100 [21] |
| Printer | EPSON PX-S160T |

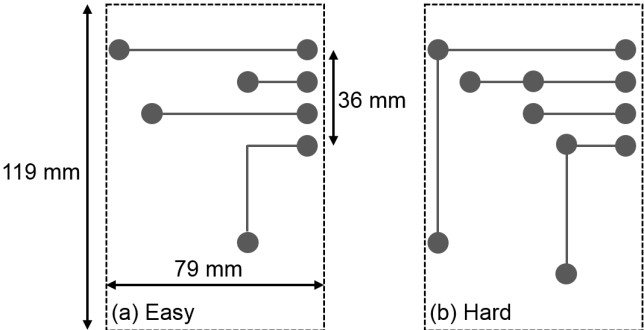

Figure 3: SheetKeys of different complexity used in experiment.

### 5.2 Apparatus

The experimental apparatuses are listed in Table 1. We used an LG Nexus 5 smartphone ($1080 \times 1920$ pixels, $61.6 \times 110$ mm$^2$ of input area, 17.5 pixels/mm resolution, portrait mode). For SheetKeys, we printed two sheets with different degrees of manual input difficulty based on our guidelines (as shown in Fig. 3). Each sheet was printed with silver nano-particle ink [20] on transparent PET film [21] by a commercially available inkjet printer [15]. The sheets were cut to $79 \times 119$ mm$^2$ and pressed onto the smartphone screen with an acrylic plate. An Android application running on the smartphone kept reading the touch events and unlocked the phone only when the set of touch events matches the secret pattern set inside the application. Again, we used a diameter of 8 mm for the discs in the *user-touch* and *touch-generation parts* and a width of 0.5 mm for the wires.

For simplicity, we divided the smartphone display into a grid so that the discs for the *touch-generation part* had an 8 mm diameter with some margins to prevent wires from conflicting. Therefore, each edge of the cells in the grid was 10 mm long, and the numbers of cells on the x- and y-axes were 5 and 8, respectively. The touch positions fell into one of the $5 \times 8 = 40$ possible cells. Thus, for **attackers**, the tap positions did not have to be exactly on the disc centers, but they did have to be in the range of $\pm 5$ mm from the disc centers. Throughout this study, when two or more taps were sensed within the range of $< 30$ ms, the system judged that they concurrently occurred and thus were a multi-tap.

### 5.3 Participants

Thirteen unpaid volunteers from our local university participated in this study. One was left-handed, and the remaining 12 were right-handed. They had normal or corrected-to-normal vision and reported no trouble with motor ability. All used their smartphones every day. For the data collected in this experiment (handedness, eyesight, health, and interview comments), ethics approval was not mandatory in our university.

### 5.4 Manual Input Difficulty

There were several choices for the manual input difficulty from the viewpoint of attackers. For example, we could test the effects of the number of single tapping operations, without including multi-tapping; as the number of required taps increases, the success rate for manual operations by attackers decreases. However, a user study

does not need to be conducted to know that increasing the total numbers of taps for both the *user-touch* and *touch-generation parts* obviously increases the operational difficulty for not only attackers but also valid users. Another choice was fixing the number of discs in the *user-touch part* and changing that in the touch-generation part by branching wires. This could also test our assumptions as described in the two guidelines. Hence, as shown in Fig. 3, we changed the degree of manual input difficulty by manipulating the number of concurrently required taps.

To fairly compare the proposed method with a conventional authentication method (a four-digit PIN), we fixed the number of touch operations to four. *Valid users* had only to slide a finger 36 mm downwards along the four discs in the *user-touch part* for both the *Easy* and *Hard* SheetKeys. *Attackers* had to perform two different operations depending on the level of manual input difficulty as follows.

- **Easy**: As the baseline for SheetKey, *Easy* required four successive single taps. We assumed that *attackers* occasionally passed through the authentication by, for example, setting four fingers above the required tap positions and then tapping in the correct order with slight delays.

- **Hard**: For the first, second, and fourth touches, two-finger multi-taps were required. The third touch required a single finger tap. To successfully achieve this, *attackers* had to switch the number of fingers for tapping from two to one and then switch back from one to two.

## 5.5 Design and Procedure

Participants sat on a chair and held the smartphone in their non-dominant hand. Because the experience of *valid users* and *attackers* (*i.e.*, using or not using SheetKeys) would not affect the other's experience, their order was not randomized. First, participants performed the *valid user* tasks using SheetKeys. Second, they performed the *attacker* tasks. For both sessions, they tried to log in 30 times for each level of difficulty. Participants were provided with the correct unlock key sequences in the *attacker* session. The order of the two levels of difficulty was randomized.

After completing all the trials, we interviewed the participants about their impressions and strategies in the experiment. This experiment took approximately 15 minutes from instruction to completion of all trials per participant. Note that the left-handed participant held the smartphone upside-down for the *valid-user* tasks and made strokes upwards on the *user-touch part*.

In total, we recorded 30 (trials) × 2 (*valid user* and *attacker*) × 2 (*task difficulty = Easy* or *Hard*) × 13 (participants) = 1,560 data points. The order of sessions was fixed from *valid user* to *attacker*, while the order of task difficulty, *Easy* and *Hard*, was randomized across participants. The 30 repetitions were for a single condition of session × difficulty, *e.g.*, a participant first tried the *Hard* sheet 30 times as a *valid user*, tried the *Easy* sheet 30 times as a *valid user*, and then played the role of the *attacker* for 30 trials per task difficulty.

## 5.6 Results

The users succeeded in 643 trials and failed in 137 with SheetKey and succeeded in 107 trials and failed in 673 without SheetKey. We analyzed the data via repeated-measures ANOVA with the dependent variables of *with/without* SheetKey, the task difficulty (*Easy* or *Hard*), and the independent variable of *success rate*.

Both the *using SheetKey* ($F_{1,12} = 324.873$, $p < 0.001$, $\eta_p^2 = 0.964$) and *task difficulty* ($F_{1,12} = 15.277$, $p < 0.01$, $\eta_p^2 = 0.560$) conditions showed significant main effects on the success rate. We also observed their interaction ($F_{1,12} = 5.700$, $p < 0.05$, $\eta_p^2 = 0.322$), as shown in Fig. 4. Pair-wise comparisons showed that the task difficulty (*Easy* or *Hard*) did not significantly affect the success

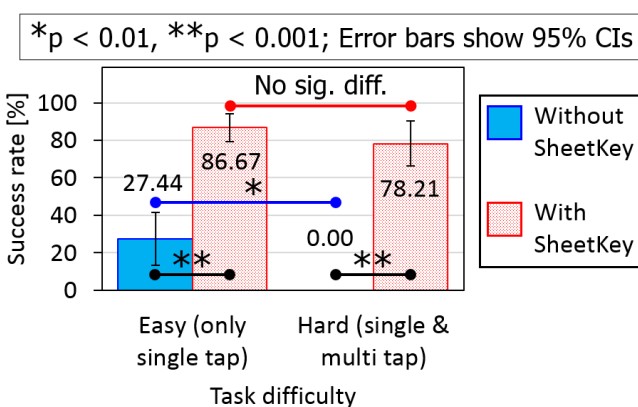

Figure 4: Average success rates for each task condition in experiment.

rate when using the SheetKey (red bars in Fig. 4), whereas the other combinations showed significant differences ($p < 0.001$–$0.01$).

## 5.7 Discussion of the User Study

In the *attacker* session, 27% of trials were successful with manual input for the *Easy* condition. As expected, participants tried to set four fingers above the required positions and then tap the touchscreen. For the *Hard* condition, this strategy, however, could not be used, because participants had to "reuse" some fingers from one required position to another. In addition, according to post-experiment interviews, switching the number of fingers for single/multi-taps made the task significantly more difficult.

In the *valid user* session, there was no significant difference between *Easy* and *Hard*, because both required the same length for finger sliding on SheetKey (36 mm). Failed trials were typically due to the sliding speed. Although we set the time limit to 0.3 s for 36 mm finger sliding, this could be refined by calibrating the typical finger sliding speed for a given distance on SheetKey. We assume that the reason behind the accuracy difference between the *Easy* and *Hard* conditions under which the *valid users* performed was the number of discs for the *touch-generation part*. Because the *Hard* sheet requires that more discs should contact the screen, stronger force was needed to robustly generate touch events.

As a comparison, because the typical error rate of tapping a circular target is 4% if a user wants to perform at a reasonable speed (rapidly and accurately aim for a target) [5, 22], the success rate for four-digit PIN input would be $0.96^4 = 84.93\%$. This is comparable with the success rate for the *valid user* sessions using SheetKeys in the *Easy* mode (86.67%). The success rate for six-digit PIN input is $0.96^6 = 78.28\%$, which is close to the success rate for *Hard* mode (78.21%). Overall, we recommend using patterns including multi-taps for maintaining a high success rate when valid users use a SheetKey and a low success rate when attackers manually input the patterns.

## 6 FURTHER IMPROVEMENTS TO SHEETKEY

On the basis of the results of the user study, we found several potential threats raised by participants. Although our fundamental goal was to increase security by using an additional tool, SheetKey would benefit from improved protection from other threats. We discuss how to deal with other possible threats below. Formal experiments done to test the validity of these alternative solutions against potential threats are included in our future work.

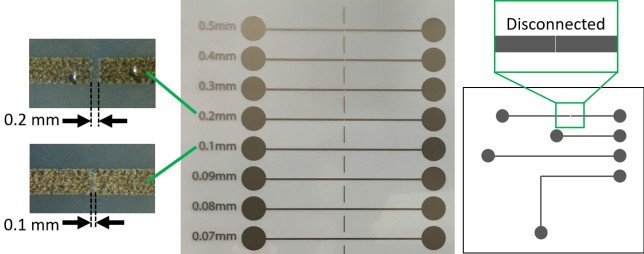

Figure 5: A wire is not connected, and touch events do not travel. 0.2 mm was the smallest gap at which touch events were not conveyed.

### 6.1 Taking Photos or Videos to Reproduce the SheetKey

#### 6.1.1 Threat

If attackers have a setup that can produce SheetKeys (conductive ink and an inkjet printer), they can produce a copy of SheetKeys after reconstructing the patterns, *e.g.*, by taking a photo or video of users inputting the patterns using a SheetKey. We propose two solutions to prevent attackers from reconstructing patterns printed on SheetKeys even if they can print SheetKeys.

#### 6.1.2 Solution 1: Using an Opaque Sheet

A straightforward method is to print a pattern on an opaque sheet and attach it on a device so that the printed side contacts the screen. Potential disadvantages of this method would be that (a) the sheet materials would be limited to opaque types and (b) valid users cannot see the discs in the *user-touch part*, which could degrade usability and the success rate.

#### 6.1.3 Solution 2: Disconnected Wires

Although the wires look like they connect two discs, some of the wires are actually split by a slight gap. On the basis of our test, a 0.2 mm gap was found to be sufficient while maintaining the look of being one single wire (Fig. 5). Such a small gap can rarely be detected 2 m away from the sheet or when taking a photo with a full high-definition (HD) camera. This method allows SheetKeys to be secure even for transparent sheets.

### 6.2 Stealing a SheetKey

#### 6.2.1 Threat

If attackers steal a SheetKey, they can illegally log in to systems/applications. In addition, they can correctly detect connected/disconnected wires by using a circuit tester, or by using SheetKey on a smartphone that feeds back the touch positions. Hence, attackers can produce counterfeits of SheetKeys. The solutions described in the previous subsection (opaque sheets and disconnected wires) are not effective against this threat.

#### 6.2.2 Solution for the screen: Adjusting a SheetKey to the Correct Symbol

With this method, a valid user has both a SheetKey and knows the corresponding symbol to which it should be attached on the screen, thus realizing two-factor authentication. The login system randomly shows many symbols such as circles and squares on the screen, and users have to adjust their SheetKey on the basis of a given rule (see Fig. 6). This rule is made by the system or programmed by users. For example, "adjust the top-left of the sheet to the top heart symbol" or "adjust the bottom-left corner of the sheet to the second bottom-most blue symbol." If attackers steal a SheetKey, they can rarely attach the sheet to the required position on a touchscreen.

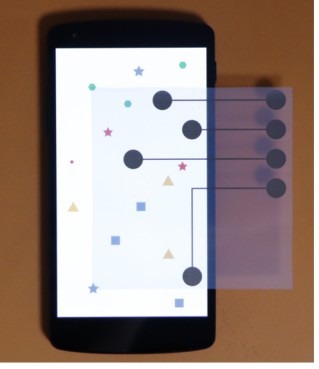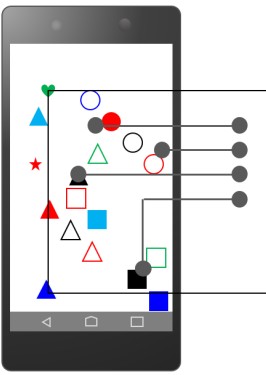

Figure 6: Adjusting the sheet to the required position on basis of displayed symbols.

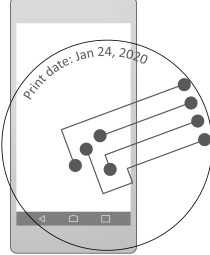

Figure 7: Adjusting the angle of the circular sheet on basis of printed information.

Even if attackers look at users' input, they will have difficulty estimating the required rule. As shown on the right of Fig. 6, from the viewpoint of attackers, there are several possible rules such as adjusting the third disc of the sheet to the leftmost black symbol or adjusting the bottom-left of the sheet to the bottommost triangle symbol. Therefore, even if attackers see users' inputting a pattern once or twice, they have to try to log in by estimating the correct rule. Security can be maintained by applying common methods such as locking an account after four invalid inputs.

#### 6.2.3 Solution for the Sheet: Using a Circular Sheet

Using a circular sheet is another way to make it more difficult for attackers to estimate the correct rule because the angle at which it is used makes this more difficult than with rectangular sheets. As a reference position, a printed disc, wire, or other information such as the date for printing can be used. In Fig. 7, a given or programmed rule for the angle such as the day ("24") of the printed date is at the top. In this example, only the angle is considered for the validation.

A potential threat is that, if an attacker sees a valid user use a SheetKey for input many times, the possibility of the attacker predicting the correct rule increases when using both random symbols and circular sheets. Hence, a limitation of these solutions is that attackers can steal a SheetKey *and* photograph/watch input be done many times. However, as an advantage of SheetKey, valid users then can invalidate the old pattern and reissue a new pattern using their own printer soon after they notice that a SheetKey has been stolen.

### 6.3 SheetKey for Alphabetical Keyboards

To realize a SheetKey for inputting passwords with higher security than 10-PIN keypads, such as a full QWERTY keyboard, there are several technical issues to overcome due to the more complex patterns for discs and wires that would be at a high density. We mentioned that the thickness of the connection wires between each conductive disc requires widths of 0.1–0.5 mm to avoid touch events

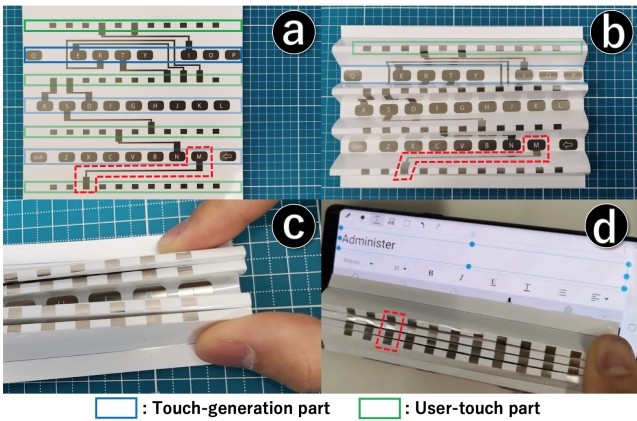

☐ : Touch-generation part   ☐ : User-touch part

Figure 8: A SheetKey for inputting a password on a software keyboard.

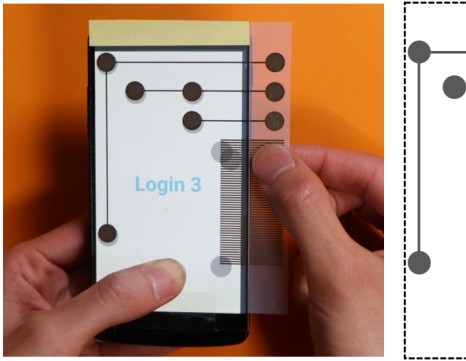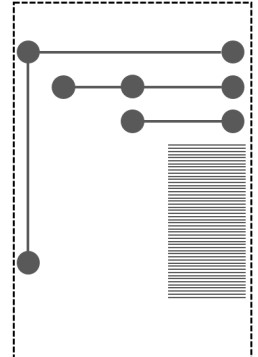

Figure 9: A SheetKey including a stripe pattern to invoke a sliding event on the touch-generation part.

being generated on an unintended portion of touchscreens in the *Measurement* section. However, it is known that touch input can be generated by multiple tightly grouped thin lines [14]. Thus, it is difficult to rewire array keys at a high density to their respective conductive discs when the layout is complicated.

To address this problem, we applied an origami technique to separate the wires and *user-touch part* from the *touch-generation part*. As shown in Fig. 8, the alphabetical-keyboard SheetKey has a two-layer structure: the *user-touch part* and the *touch-generation part*. These two parts are printed in alternate order on a PET sheet (Fig. 8-a). The *user-touch part* consists of a group of four separate conductive rectangles. Users line up the rectangle parts by bending the sheet, and a touch input is generated by sliding a finger along the top of the origami structure (Fig. 8-b,c,d).

Each conductive rectangle of the *user-touch part* is $3.0 \times 4.0$ mm$^2$ and connected to a corresponding disc of the *touch-generation part*. When the user slides their finger across the lined-up rectangles of the *user-touch part*, only those conductive rectangles that are connected to the *touch-generation part* are activated. When the user slides their finger rightwards, the conductive rectangles framed in the red are activated in a preprogrammed order (Fig. 8-d). In the example of Fig. 8, when the user touches the third *user-touch part* from the left (Fig. 8-d), the "M" key, which is connected to the bottom conductive rectangle (8-a,b), is activated.

The conductive discs of the *touch-generation part* have an elliptical shape measuring 10.0 mm by 6.0 mm, and the spacing between each disc is 3.0 mm. According to the result of a preliminary study, this size is enough to generate a touch input. In this paper, we realized a full QWERTY keyboard with SheetKey using a Samsung Galaxy Note 8 (20.7 pixels/mm resolution, landscape mode) with a Google Keyboard application (each key size was $12.5 \times 6.5$ mm$^2$). The key size of this software keyboard depends on the display size and resolution of the smartphone. Thus, an alphabetical keyboard SheetKey pattern requires designing to fit the size and position of software keys.

### 6.4 SheetKey Invoking a Sliding Event

We also applied a sliding motion via stripe patterns [14] to SheetKey (Fig. 9). Sliding a finger on the stripe requires more difficult finger movement than that in a tapping task (cf. a drag-and-dropping task [23] or a steering law task [3]). In addition, sliding operations require a certain length of time, and thus, the strategy of setting fingers above the required position and tapping cannot be applied to manual input by attackers. However, this technique is difficult to evaluate on the basis of existing methods such as entropy. Thus, a fair indicator of validation needs to be created.

## 7 DISCUSSION

### 7.1 Security

The entropy of the pattern depends on the screen size. The cell area of 10 mm$^2$ used in our user study is needed so that one tap point is not affected by the other nearby points. For instance, we explain the pattern entropy of SheetKey for the Nexus 5 smartphone by comparing it with PIN input with 10 key code entropy. Again, 40 points can be located on this smartphone ($61.1 \times 110$ mm$^2$ screen size) to reserve margins.

In the case of four single taps (see Fig. 3-left), because one disc in the *touch-generation part* can be used multiple times, the complexity is $(_{40}C_1)^4 = 2,560,000$ possible patterns. This is about 2.5 times as complicated as inputting a 6-digit PIN code using a combination of 10 numbers ($10^6 = 1,000,000$). When the multi-tap condition is added, the complexity ($(_{40}C_2 + _{40}C_1)^4 = 4.52 \times 10^{11}$) increases by about 100,000 times and has the same complexity as about 11-digit input using 10 numeric combinations.

Although the user swipes the discs of the *user-touch part* sequentially, the touch events on the screen should be generated at random for more security. However, connecting the disc arrays in the *user-touch part* and *touch-generation part* is not a naive problem. To design complex (*i.e.*, higher security) disc layouts easily, a double-sided print [30] is employed in order to keep patterns from crossing.

We also propose a unique method for automatically generating a wiring pattern between the user-touch points and touch-generation points based on the idea of Levenshtein distance [18]. The number of via holes required for connecting the front and rear surfaces is calculated as the Levenshtein distance of two sequences corresponding to the touch sequence of the discs of the *user-touch part* and *touch-generation part*. The Levenshtein distance is calculated with a $O(nm)$ time complexity, which is a practical calculation order, for the number of input points $m$ and the number of points of the touch generation part $n$, and the number of via holes is less than $\max(m, n)$. Thus, all of the discs in the *touch-generation part* can be used, and the number of user-touch points does not sacrifice the entropy of SheetKey.

Judging from the result of calculating code entropy, we can conclude that the complexity of pattern input can be sufficiently secured even when processing only four discs in the *user-touch part*. Furthermore, security can be strengthened when picking up a SheetKey by making it necessary to check the relative position between the sheet and an application.

### 7.2 Limitations and Future Work

One of our concerns include the fact that SheetKeys are potentially fragile due to the exposed conductive ink. However, Kawahara et

al.'s experiment showed that the resistance of printed conductive ink increased by about 15% after 7 months [15]. In the current use case, the longest wiring of SheetKey is about 10 cm, so it is not a problem to use it even if the resistance value is slightly increased. It is also possible that the printing surface may deteriorate due to repeated touch operations by the user, but this can be dealt with by coating the surface with thin tape.

Although we demonstrated the effectiveness of SheetKey against manual input attacking, our discussions are somewhat limited owing to the experimental conditions such as the number of discs for the SheetKeys. We are also interested in other disc layouts in the *touch-generation part*. For example, if we set the four discs in a linear manner, attackers might achieve a higher success rate. Conducting user studies to evaluate such conditions is future work and will contribute well to improving our method.

When creating SheetKeys on Adobe Illustrator, the time needed depends on the complexity of the sheet and the designer's experience. For example, the simple patterns shown in Fig. 3, which use only circles and lines, took an experienced user less than 5 min. However, if exploring more complicated wiring such as shown in Fig. 8, it could take more than an hour. Although we designed our SheetKeys somewhat heuristically, developing end-user support tools will contribute to the deployment of SheetKey in the future.

In this paper, we mainly discussed the technical and theoretical aspects of SheetKey. From the viewpoint of usability, however, we came up with possible problems with SheetKey. First is its mobility, because users have to carry a SheetKey with their smart devices or bring it to a secure room. However, unlike other methods that require additional items such as hardware tokens, carrying a light-weight sheet to achieve quite higher security than four- or six-digit PINs would not be too burdensome. Easily trashing and reissuing a SheetKey would also helpful for maintaining the security level, the way a web service locks a user account and unlocks it via email authentication. Second is the method for attachment. One choice is, as we did in the *Measurement* section, attaching a SheetKey onto a touchscreen with double-sided tape. Another way is using a smartphone cover to press a SheetKey to a touchscreen. A cover with a certain thickness (~2 mm or more) would not invoke touch events, and thus, many consumer covers can be used for it.

Evaluating the time cost and effort to align a SheetKey to the appropriate position on the screen is also important for usability. In addition to the positioning time, as the number of symbols increases, the time taken to determine the correct alignment increases. Therefore, as in other secure methods for end-users, SheetKey also has a tradeoff between usability and security.

## 8 CONCLUSION

We proposed SheetKey, a sheet that generates a complicated touch input pattern printed with conductive ink for secure authentication against shoulder surfing. We showed the optimal diameter for the discs used to generate touch events (8 mm) and wire width (0.1–0.5 mm). The results of our user study demonstrated that *attackers* had great difficulty logging in by manually inputting patterns, while *valid users* easily logged in using SheetKeys. We also discussed and implemented potential improvements to SheetKey. Comparison with conventional PIN inputs showed that SheetKey is sufficiently complex. Although we focused on technical evaluations in our user studies, we will also investigate the effectiveness and limitations through observation using SheetKeys.

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
