# OpenReview forum: "SheetKey: Generating Touch Events by a Pattern Printed with Conductive Ink for User Authentication"
_graphicsinterface.org/Graphics_Interface/2020/Conference — GI 2020_

### Official Review · AnonReviewer2 · 2020-02-11
**Review for SheetKey: Generating Touch Events by a Pattern Printed with Conductive Ink for User Authentication**

**Rating:** 5
**Confidence:** 3

**Review:**

This paper contributes to the development of SheetKey, a tangible electronics-based system used for password/pin authentication. The paper includes a discussion of the implementation of SheetKey and a preliminary evaluation of 13 participants acting as valid users who use sheetkey for password authentication and as attackers who try to guess the password.  The paper also discussed several future directions for improving the system.

Overall, the paper is well written and easy to read. The development of a relatively accessible solution i.e., printing sheetkeys using a regular printer is promising.  The current draft of the paper seems to raise more questions than it answers, which I think is fine for exploratory research, but that said, I found it difficult to fully understand the contributions of this work and below I elaborate on my main concerns.

(A) Technical implementation: the paper mentions that the main contribution of this work is the technical development of SheetKey. I think there is some novelty in the use of conductive ink, however, I am not convinced that the silver nano-particle ink is as widely accessible as the paper argues to emphasize the benefit for mass production. Also, I found it difficult to understand what the technical differences are compared to the CipherCard project cited in the paper i.e., what additional features does this technique offer? The paper mentions the use of sheetkeys on alphabetical keyboards but I missed seeing any evaluation for that.

(B) Evaluation: the basic usability testing makes sense, however, because the paper claims that one of the novel aspects of this method is that it can be easily reproduced by people, I was hoping to actually see participants go through an end-to-end evaluation i.e., set a password, print the sheetkey, and use it for authentication.  Also, the paper indicates that 27% of the attackers were successful in identifying the pattern. This seems like a relatively high number for critical tasks concerned with security, making me wonder what then is the benefit of this approach?

(C) User Experiences: a majority of the user experiences related questions were indicated as future work. For me, this is slightly problematic for an HCI publication.  While the engineering efforts are extremely important, it is difficult to fully assess the benefits of the work without a sense of how and when would people use such techniques.

In summary, I think the work is interesting, but as currently presented it is difficult to fully assess what the contribution of this work is. Just based on reading the paper, I think a short paper or a poster might be a better fit for this work.

---

### Official Review · AnonReviewer1 · 2020-02-11
**Interesting idea, could be further explored**

**Rating:** 6
**Confidence:** 4

**Review:**

Overall, this is a clever and interesting idea which takes the idea of a physical key and applies it to a capacitive touchscreen. The analogy to a key is quite apt - the proposed devices produce inputs that have combinatorially many possibilities, that are physically distinct and need to be physically protected, and whose inputs are hard to forge by hand (without simply designing a replacement key).

The idea of printing conductive sheets to interact with a screen is not new (as evidenced by the related work, and also [A] below), but this is a nice physical-key based technique for authentication that could complement existing capacitive-biometric techniques (e.g. [B]) as a primary or secondary authentication factor.

I have a few concerns about the paper. The accuracy as demonstrated in the paper seems somewhat low (~80%), which could significantly impair usability, and it's not clear why the accuracy is low (touch sensing on the screen itself is in excess of 99% accurate). Details on the recognition algorithm are not clear; I wonder if the authors are matching on the absolute positions of the touch points, or on their positions relative to other touch events.

The keys are large, a bit unwieldy, and seem potentially fragile due to the exposed conductive ink, which might make them hard to carry around; this issue makes them much less attractive than e.g. smartcards or fobs for authentication. Some ideas on how they can be made more robust and easy to use, and not simply "trashed and reissued" when they fail (which would require security procedures at many institutions equivalent to reissuing a normal key) would be strongly welcome.

Finally, the user has to hold the key to the screen and thereby inject their own touches onto the screen, while also being somewhat unergonomic. Some ideas are given but these seem to assume that users will carry around additional hardware, which might be onerous.

In terms of the evaluation, I would have liked to see some more details from the fabrication side of the project - how easy is it to design these keys, how much it costs to print them and how long it takes, etc. There is some anecdotal information in the introduction but no hard numbers.

The keyboard is a very interesting and cute idea! I like that it can be used to input a secure password/PIN which users do not have to memorize.

This paper is also missing a few references:

[A] Yuntao Wang, Jianyu Zhou, Hanchuan Li, Tengxiang Zhang, Minxuan Gao, Zhuolin Cheng, Chun Yu, Shwetak Patel, and Yuanchun Shi. 2019. FlexTouch: Enabling Large-Scale Interaction Sensing Beyond Touchscreens Using Flexible and Conductive Materials. Proc. ACM Interact. Mob. Wearable Ubiquitous Technol. 3, 3, Article 109 (September 2019), 20 pages. DOI:https://doi.org/10.1145/3351267
[B] Anhong Guo, Robert Xiao, and Chris Harrison. 2015. CapAuth: Identifying and Differentiating User Handprints on Commodity Capacitive Touchscreens. In Proceedings of the 2015 International Conference on Interactive Tabletops & Surfaces (ITS ’15). Association for Computing Machinery, New York, NY, USA, 59–62. DOI:https://doi.org/10.1145/2817721.2817722

[A] discusses a method for fabricating flexible films for capacitive interaction in the same fashion as described here, and proposes a number of use cases.
[B] describes a technique which similarly uses the touchscreen to authenticate users, using their palmprint.

Overall, I think this paper is above the bar for publication to GI, pending a better discussion of the current limitations and a more thorough comparison with related work.

---

### Official Review · AnonReviewer3 · 2020-02-12
**New prototype with potential**

**Rating:** 8
**Confidence:** 4

**Review:**

The authors present Sheetkey, a sheet with conductive patterns that allow the users to quickly enter complicated touch pattern passwords (<0.3s). I like the idea of this prototype, it’s simple, straightforward, and powerful. The application is appropriate. Of course, the big downside to this method is the necessity for yet again another thing for users to carry, and that it can easily be stolen. The authors do discuss this in the paper.

Introduction lists a number of future work to clarify where the paper stands. This is an interesting writing technique, but it leaves me with more questions than I had before, mainly why weren’t some of these done here? They seem very basic and critical to evaluating the usability of the method. None the less, the effect is that the paper’s aim and scope is clear and I very much appreciate that.

Similarly, I think that the paper is well organized with regards to its content: there are small studies, but the improvement sections clearly outlines issues and how to fix them, it’s not left to future work, which strengthens the work.

One argument that is weaker is that this sheet is easily produced with a consumer printer and ink – the ink is specialized, so it’s not really something easy to produce. However, I understand that it might be in comparison to complex security key systems.


Study design:
-	Were all 30 tasks for each level of difficulty different, or were there repetitions?
-	Was the randomization done per block (easy-difficult or difficulty-easy) or were the easy and hard tasks mixed?

A few more details would enhance the paper:
-	Add the width of the disk in preliminary study 1
-	Can you have an overlapping figure with the sheet and the phone, as to compare the sizes? I understand why the sheetkey must be larger but not why it must be taller than the screen.
-	Please add details regarding compensation of participants and whether this study received institutional ethics approval.
-	“interviewed their impressions” -> “interviewed THEM ABOUT  their impressions”

Overall, I recommend the acceptance of this work, it’s interesting and appropriate.

---

### Meta-Review · Area_Chair1 · 2020-02-17

**Recommendation:** Accept
**Confidence:** 5

**Metareview:**

Three expert reviewers reviewed this paper. R2 noted that the paper could have gone further with the evaluation, to test the full usability of the system, and was concerned that the contribution of the paper was unclear. R1 is a bit more positive, with some comments on the technical aspects of the work. Finally, R3 is the most positive, and is strongly in favor of acceptance, with some requests for additional details.

Overall, I believe the paper is above the bar for publication to GI, based on the originality and quality of this work. While the evaluation does not go very far into the usability aspects, some of which are left to future work, the system as a proof of concept more than meets the bar (a completed implementation, an evaluation of the technical aspects, and some sample application areas).

In revisions to the paper, I ask that the authors resolve the following issues prior to acceptance:
- (all Rs) Add some details substantiating the claim of the conductive ink as being easily/widely accessible (e.g. cost, availability), or remove this claim
- (R1, R2) Add some details on the fabrication process, e.g. fabrication design and print time, costs
- (R3) Add several details about the evaluation to the paper: details on study design (type/randomization of tasks, compensation of participants, ethics approval), details on sheetkey (size of disk)

The authors should also strongly consider the following suggestions:
- (R1) Consider including a few additional references in related work
- (R1) Consider discussing reasons why the accuracy is low (~80%)
- (R3) If space allows, add a figure of the SheetKey overlaid on the phone to show scale

---

### Decision · Program_Chairs · 2020-02-18

Accept